# Comparing feedback and spatial approaches to advance ecosystem-based fisheries management in a changing Antarctic

**Emily S. Klein**[1,2¤]*, **George M. Watters**[1]

**1** Antarctic Ecosystem Research Division, Southwest Fisheries Science Center, National Marine Fisheries Service, National Oceanic and Atmospheric Administration, La Jolla, California, United States of America, **2** Farallon Institute, Petaluma, California, United States of America

¤ Current address: Frederick S. Pardee Center for the Study of the Longer-Range Future, Boston University, Boston, Massachusetts, United States of America
* emily.klein04@gmail.com

**Data Availability Statement:** Data are contained within the manuscript or are available in published papers cited therein, and all code and input files are available online at https://github.com/EmilyKlein/KPFM2.

## Abstract

To implement ecosystem-based approaches to fisheries management, decision makers need insight on the potential costs and benefits of the policy options available to them. In the Southern Ocean, two such options for addressing trade-offs between krill-dependent predators and the krill fishery include "feedback management" (FBM) strategies and marine protected areas (MPAs); in theory, the first adjusts to change, while the later is robust to change. We compared two possible FBM options to a proposed MPA in the Antarctic Peninsula and Scotia Sea given a changing climate. One of our feedback options, based on the density of Antarctic krill (*Euphasia superba*), projected modest increases in the abundances of some populations of krill predators, whereas outcomes from our second FBM option, based on changes in the abundances of penguins, were more mixed, with some areas projecting predator population declines. The MPA resulted in greater increases in some, but not all, predator populations than either feedback strategy. We conclude that these differing outcomes relate to the ways the options separate fishing and predator foraging, either by continually shifting the spatial distribution of fishing away from potentially vulnerable populations (FBM) or by permanently closing areas to fishing (the MPA). For the krill fishery, we show that total catches could be maintained using an FBM approach or slightly increased with the MPA, but the fishery would be forced to adjust fishing locations and sometimes fish in areas of relatively low krill density–both potentially significant costs. Our work demonstrates the potential to shift, rather than avoid, ecological risks and the likely costs of fishing, indicating trade-offs for decision makers to consider.

## Introduction

Implementing an ecosystem approach to the management of fisheries, and balancing ecological and human needs, is widely acknowledged as critical (*e.g.* [1]), especially in a changing climate (*e.g.* [2]). A vital element of ecosystem-based approaches, across all variants (e.g.

**Funding:** EK was supported by funding from the Pew Charitable Trusts, contract ID #31740. This funder had no role in study design, data collection and analysis, or preparation of the manuscript. Publication under peer review was a requirement of this funding source, but the funder did not take part in deciding where this manuscript would be submitted or any part of the submission process. There was no additional external funding received for this study.

**Competing interests:** The authors have declared that no competing interests exist.

ecosystem-based management, ecosystem-based fisheries management, etc.), is dealing effectively with uncertainty and changing conditions (*e.g.* [2, 3]), especially given the potential impacts of climate change. How do we facilitate sustainable use of ecosystems while addressing an unclear future?

The Southern Ocean is an ecologically and economically rich region [4, 5] where the consequences of climate change, such as higher temperatures and declining sea-ice extent, are already being observed (*e.g.*, [6–8]). Here, the Commission for the Conservation of Antarctic Marine Living Resources (CCAMLR or the Commission) is responsible for ensuring that the structure and function of marine ecosystems are maintained as humans utilize various ecosystem services [9, 10]. The Commission also recognizes the need to mitigate current and future climate-change effects (*e.g.*, [11]). One approach the Commission has considered to meet its conservation objectives given variability and change is "feedback management" (FBM) of the Antarctic krill (*Euphasia superba*, hereafter krill) fishery [12, 13]. CCAMLR defines FBM for the Antarctic krill fishery as using "decision rules to adjust selected activities (distribution and level of krill catch and/or research) in response to the state of monitored indicators" [12].

Feedback management strategies have a long history of discussion within CCAMLR (*e.g.*, [12, 14–17]) and in the wider literature (*e.g.*, [18–20]). Here, we refer to FBM as CCAMLR defines it, wherein a decision rule is applied to adjust the catch levels and their spatial distribution based on the state of a monitored indicator. The decision rule itself is fixed and is not modified, but, since the prescribed response changes depending on the value of the indicator(s), FBM can react to changing conditions. The decision rule in FBM can also readily use indicators based on the status of, or changes in, non-target species, thus potentially allowing management to be more explicitly ecosystem-based [18]. Therefore, feedback management strategies are a possible approach by which decision makers can maintain an ecosystem focus while also addressing uncertainties around future climate change by regularly adjusting to changing conditions. We note that FBM is not adaptive management, as nothing is learned via the process and the decision rule is not updated [21], nor is it traditional fisheries management, as it does not require further information or modeling beyond the indicator [17, 22, 23].

Of course, FBM is only one tool in the proverbial toolkit; the effectiveness of such approaches needs to be assessed prior to implementation and against other options (*e.g.*, [17, 18, 21]). Marine protected areas (MPAs), wherein extractive uses like fishing are limited or prohibited, are an alternative to FBM. Protected areas may also be valuable for addressing uncertainty [24–26] and applying an ecosystem approach to management (*e.g.* [27]). While FBM strategies *adjust* to change, effective and carefully designed MPAs are a strategy that can potentially be *robust* to change [28, 29].

Both FBM and MPAs have been prioritized for policy consideration in the Southern Ocean, and CCAMLR has established these priorities given the current and ongoing impacts of climate change and the need for management despite inevitable uncertainty. In 2011, the Commission adopted a set of objectives to be met by MPAs throughout the Southern Ocean (Conservation Measure 91–04) [30]. These objectives nominally parallel those of FBM as both ultimately intend to achieve the overarching aims of Article II of the Convention that established CCAMLR, namely the conservation of marine living resources [31–33]. The Commission is currently wrestling with the concept of "harmonizing" FBM with the establishment of an MPA in an important krill-fishing area in the Scotia Sea and Antarctic Peninsula region.

Both FBM and MPAs pose challenges in addition to their benefits, and both can be difficult to implement. To encourage their implementation, it is critical that the potential outcomes of these approaches are assessed, especially as the climate changes over the long term. Dynamic ecosystem models are useful tools to conduct such assessments [34] and may be especially valuable for evaluating the performance of MPAs [35]. These simulation models are

increasingly used to support management decisions [36]. In the Southern Ocean, the Commission advocates using ecosystem models to expedite the delivery of scientific advice on FBM strategies [12], and such models are intended to and have been used to evaluate MPAs [37, 38].

Here, we utilized a dynamic ecosystem model [39] to support decision making in the Southern Ocean by comparing two FBM strategies against one another and against an MPA given the impact of climate warming on krill growth [40, 41]. For the two FBM strategies, we projected outcomes in which spatially resolved catch limits for "small scale management units" (SSMUs, [42]) were periodically updated based on indicators of either (1) krill density or (2) changes in penguin abundance. We also projected the outcomes of an MPA previously shown in other modeled scenarios to have ecological benefits and to be capable of buffering possible consequences of climate change [38]. Our comparison provides guidance on addressing an uncertain future with either FBM, an approach aimed at adjusting to change, or a potential MPA, which may be robust to change. Our findings are directly relevant to CCAMLR and active conversations therein, but also hold broad implications for ecosystem-based management in an uncertain future.

## Methods

To evaluate two FBM strategies and a candidate MPA in the Antarctic Peninsula and Scotia Sea, we employed the Krill-Predator-Fishery Model (KPFM2, [39]). KPFM2 has previously been used to develop scientific advice on krill-fishery management (*e.g.*, [38, 39, 41]). The model is minimally realistic, i.e. it focuses on the specific subset of a complex and coupled system that is deemed most relevant to the questions at hand (as in Plagányi et al. [43]). Here, as with previous use of KPFM2, those questions revolve around trade-offs between development and expansion of the international krill fishery and conservation of krill-dependent predators as they compete for finite krill resources.

Watters et al. [39] described KPFM2 in detail, with sensitivity analysis further provided by Hill & Matthews [44]. The model is currently parameterized with a seasonal time step (summer and winter) and projects outcomes for the fishery alongside those for four krill-dependent predator groups, penguins, seals, whales, and fish (Table 1). While both FBM strategies we address here involve a single indicator species, it is important to assess possible impacts on other species, which may also signal the potential for broader, system-level outcomes. We focus on penguins (an indicator in one strategy) and seals (not an indicator in either) to keep main text figures manageable, with results for whales and fish provided in (S1 File). We employed the ecological and model structure from Watters et al. [39], with the updates and MPA delineation outlined in [38]. Briefly, the dynamics of krill and each predator group are described by delay-difference equations (S1 File) in which temporal trends in abundance are recursive (e.g. the abundance of seals in one SSMU at the current time step depends on the abundance of seals in that same SSMU during the previous time step). Krill predators are modeled as resident populations in one SSMU (Table 1), with each population foraging across multiple SSMUs, as defined by recent tracking data collected during the breeding (summer) and non-breeding (winter) seasons. The post-larval biomass of krill in each SSMU is estimated at the beginning of each time step, and is determined by stochastic recruitment and area-specific mortality and movement. Competition arises when krill biomass is insufficient to satisfy the combined demand of predators and the fishery. Additional information is provided in the (S1 File), and input data and code are available open access and online [45]; however, such data and code repositories are not to serve as a comprehensive explanation of the ecosystem model and its use. For additional details on the model and its rationale, we strongly encourage interested readers to refer to Watters et al. [39].

**Table 1. Species composition of krill predator groups and where they are modeled as resident by small scale management unit (SSMU, Fig 1).**

| Predator group | Modeled as resident in SSMU | | | | | | | | | | | | | | |
|---|---|---|---|---|---|---|---|---|---|---|---|---|---|---|---|
| Common name (*Species name*) | 1 | 2 | 3 | 4 | 5 | 6 | 7 | 8 | 9 | 10 | 11 | 12 | 13 | 14 | 15 |
| Penguins | | X | X | X | X | X | X | X | | X | X | X | | X | X |
| Adélie penguin (*Pygoscelis adeliae*) | | | | | | | | | | | | | | | |
| gentoo penguin (*Pygoscelis papua*) | | | | | | | | | | | | | | | |
| chinstrap penguin (*Pygoscelis antarctica*) | | | | | | | | | | | | | | | |
| macaroni penguin (*Eudyptes chrysolophus*) | | | | | | | | | | | | | | | |
| Seals | | | X | X | | | X | | | | | | | X | X |
| Antarctic fur seal (*Arctocehpalus gazella*) | | | | | | | | | | | | | | | |
| Whales | X | | | | | | | | X | | | | | | |
| fin whale (*Balaenoptera physalus*) | | | | | | | | | | | | | | | |
| humpback whale (*Megaptera novaeangeliae*) | | | | | | | | | | | | | | | |
| Minke whale (*Balaenoptera bonaerensis*) | | | | | | | | | | | | | | | |
| southern right whale (*Eubalaena australis*) | | | | | | | | | | | | | | | |
| blue whale (*Balaenoptera musculus*) | | | | | | | | | | | | | | | |
| Sei whale (*Balaenoptera borealis*) | | | | | | | | | | | | | | | |
| Fish | X | X | X | X | X | X | X | X | X | X | X | X | X | X | X |
| Nichol's lanternfish (*Gymnoscopelus nicholsi*) | | | | | | | | | | | | | | | |
| Antarctic lanternfish (*Electrona antarctica*) | | | | | | | | | | | | | | | |
| Macherel icefish (*Champsocephalus gunnari*) | | | | | | | | | | | | | | | |

Table adapted from Watters et al. [39].

We leveraged earlier implementations of KPFM2, including its "reference set" of four parameterizations that bracket key uncertainties about rates of area-specific krill movement between SSMUs (no movement and movement as passive drifters) and relationships between krill biomass and the effective numbers of breeding predators (hyperstable and linear) [39]. We used parameters values from previous implementations of KPFM2 (i.e. Watters et al. [39] with updates in [41, 38]), only adjusting for the FBM and the MPA scenarios considered here as described below. Spatially, the model arena covers three of the CCAMLR statistical subareas in the Atlantic Sector of the Southern Ocean, Subareas 48.1, 48.2, and 48.3 (Fig 1). These sub-areas are further subdivided into the SSMUs to better address the ecosystem impacts of krill fishing by providing a management mechanism to spatially distribute catches [42]. We report model outcomes aggregated across the model's entire spatial arena (e.g. relative change in total number of seals in the entire arena) and at the SSMU scale (e.g. change in number of seals by SSMU).

We modeled the krill fishery given a plausible future wherein the fishery is fully developed, i.e. the modeled fishery is allowed to take krill up to the total precautionary catch limit established by CCAMLR. Currently, catch limits for the krill fishery are much lower than those modeled here (about 0.01 times the biomass of krill in our study arena), however the Commission desires a spatial management strategy which successfully mitigates risks to krill predators and will ultimately allow catch limits to be increased and the fishery to fully develop [46]. Either FBM or an MPA could potentially constitute such a strategy, so we chose to model the fully developed fishery. Given this decision, we computed catch limits as the products of (1) the initial krill biomass across the model arena; (2) the harvest rate that CCAMLR used to establish the current total precautionary catch limit for krill in our study area (0.093); and (3) proportions that distribute the overall catch limit among SSMUs (and see Model

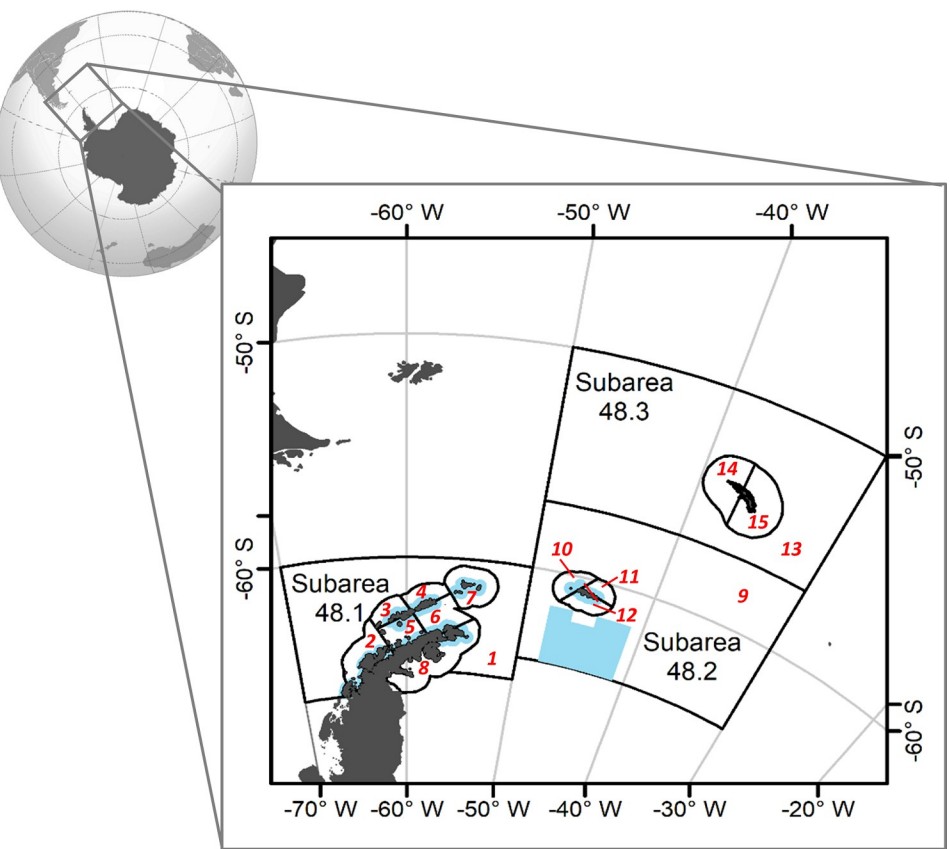

**Fig 1. Spatial structure of the ecosystem model.** Subareas 48.1, 48.2, and 48.3 are labelled, and within these, small-scale management units (SSMUs; [42]) are also outlined, as well as labeled in red; the modeled MPA is in light blue.

Implementation of FBM, below). We set the proportions in (3) equal to the average spatial and seasonal distributions of catches taken by the krill fishery from 2009 to 2017 [47].

To consider climate change in our comparison of two FBM strategies and an MPA, we followed Klein et al. [41] and simulated the potential effect of changing water temperatures on the gross growth potential (GGP) of krill [40]. We adjusted the average mass of individual krill over 100 years using the same method as in our earlier work, but, to avoid doubling the number of scenarios modeled, only included trends in krill growth given temperature changes under the Representation Control Pathway (RCP) 8.5 [48]. This pathway assumes no future action is taken to mitigate climate change.

## Model implementation of FBM

KPFM2 was designed to evaluate management strategies that adjust both the level and distribution of catch limits for krill, including simple FBM strategies. We updated and applied that latter functionality here–our only difference from the model implemented in previous work (*e.g.* [38, 39, 41]). To model how the overall catch limit is distributed among SSMUs under FBM, we developed two new "fishing options" in KPFM2 (*sensu* [49, 50]). For these FBM fishing options, we distributed overall catch limits based on SSMU-specific estimates of (1) the density of krill (g·m$^{-2}$), hereafter "FBM-Krill", or (2) changes in the abundance of breeding penguins, "FBM-Pengs". The first option, FBM-Krill, is motivated by a recommendation from Hill et al. [51], and there has been substantial recent attention on endeavors to survey krill with fishing

vessels (*e.g.*, [52]). We developed FBM-Pengs based on the considerable interest in and development of efforts to estimate the abundance of breeding penguins from remotely sensed imagery as an indicator of ecosystem health (*e.g.*, [53]). Therefore, of the myriad possible FBM approaches and indicators, such as changes in an environmental variable or abundance of other species in Antarctica, both of the FBM strategies simulated here have interest and support among a number of Southern Ocean stakeholders.

We executed the FBM fishing options using the existing reassessment framework in KPFM2 [49, 54]. In this framework, the distribution of catch limits among SSMUs is reassessed and adjusted at regular intervals during model simulations [49]. For such reassessments, the model "samples" previously simulated data at user-specified intervals and updates the catch-limit distribution as needed. We redistributed catch limits among SSMUs via this reassessment every five years. In FBM-Krill, the proportional distribution of the overall catch limit was based on the relative distribution of krill density among SSMUS; SSMUs with the highest density of krill were allocated the highest catch limits. For FBM-Pengs, the proportional distribution of the overall catch limit was based on changes in the abundance of breeding penguins, with the highest catch limits allocated to the SSMU with the greatest increase (or smallest decrease) in penguin abundance. Also in FBM-Pengs, the SSMU with the greatest decline in abundance during the reassessment interval was closed to fishing during the following interval.

In KPFM2, the overall catch limit is proportionally distributed among SSMUs following equation A.8 in Watters et al. [39];

$$\Theta_i = \frac{(B_0 * \gamma * p'_i)}{\overline{w_i}}$$

where $\Theta_i$ is the catch limit allocated to SSMU $i$, $B_0$ the initial biomass of krill, $\gamma$ the harvest rate used to compute the overall catch limit from the initial biomass, $p'_i$ the proportional distribution of the overall catch limit to SSMU $i$ (also called an "allocation fraction"), and $\overline{w_i}$ is the average mass (g) of krill in SSMU $i$. We used $p'_i$ to regularly update how the overall catch limit was distributed among SSMUs via the FBM fishing options by adjusting $p'_i$ in response to monitored indicators, either krill density (g·m$^{-2}$) for FBM-Krill or changes in the abundance of breeding penguin for FBM-Pengs.

With FBM-Krill, we "sampled" krill density, $d_i$, during the summer season in a reassessment year (timestep $t$).

$$d_i = \frac{K_{i,t}}{A_i}$$

$K_{i,t}$ is the abundance of krill in SSMU $i$ at time $t$, and $A_i$ is the area of SSMU $i$. We rescaled these density estimates to occur in the interval [0.0, 1.0] for use as $p'_i$.

$$p'_i = \frac{d_i}{\sum d_i}$$

For FBM-Pengs, we sampled the abundance of breeding penguins in each SSMU during summer of a reassessment year, time $t$, and compared it with abundance during the summer in the previous assessment year, time $t$-$10$ (KPFM2 is currently parameterized with two seasons per year and we reassessed the status of our indicators every five years). We then computed the difference between these samples ($\Delta P_i$);

$$\Delta P_i = P_{i,t} - P_{i,t-10}$$

$P_{i,t}$ is the abundance of penguins in SSMU $i$ and at time $t$. As changes in abundance can be positive or negative, we remapped the SSMU-specific changes in abundance to be $\geq 0$;

$$p_i = \Delta P_i + (\alpha \times |\min(\Delta \boldsymbol{P})|)$$

$\Delta \boldsymbol{P}$ is the vector of changes in penguin abundance that includes all SSMUs, and $\alpha$ is a scalar that determines the sensitivity of allocation fractions to changes in penguin abundance. Here, we set $\alpha$ to 1.0, which also closed the SSMU with the greatest decline in penguin abundance to fishing until the next reassessment. Finally, we rescaled each $p_i$ to occur in the interval [0.0, 1.0];

$$p\prime_i = \frac{p_i}{\sum p_i}$$

## Model implementation of the MPA

To implement an MPA in KPFM2, we used boundaries previously proposed to CCAMLR by the Delegations of Argentina and Chile (Fig 1) [55, 56]. The MPA is confined to Planning Domain 1, i.e. Subareas 48.1 and 48.2, and is therefore referred to here and by CCAMLR as the Domain 1 MPA, or "D1MPA" [57]. An MPA already exists in Planning Domain 1, the South Orkney Islands Southern Shelf MPA [58], and we combined this with the D1MPA for our analysis. For simplicity, all areas within the MPA are treated as "no-take" areas in KPFM2, i.e. they are closed to fishing, and all areas outside the MPA are open to fishing. We implemented the MPA and relevant model parameters as in [38], but here we included the effects of warming temperatures on krill growth (from [41] as discussed above).

We previously modeled three alternative reallocations of fishing displaced by an MPA [38], but, for simplicity, only include one here. We redistributed displaced catches across all open areas in the model arena, in proportion to the recent spatial and seasonal distribution of krill catches by SSMU (2009–2017, [47]). Therefore, while it is difficult to predict how catches will reallocate in reality, the alternative used here is informed by recent, real-world fishing patterns. Also, in the model, this distribution previously projected the greatest benefits (highest catches) and lowest costs (lowest probability of fishing in areas of low krill density) for the fishery in the model [38].

## Scenario assessment

We ran five scenarios using the KPFM2 model: FBM-Krill, FBM-Pengs, and the MPA, as well as "No FBM" and "No MPA" 'base case' scenarios. The base case scenarios were parameterized identically to their analogous FBM or MPA scenarios except they did not include the management strategy, i.e. FBM or an MPA. For all scenarios, we projected outcomes to the end of the 21st century and ran 1001 Monte Carlo trials (with random variations in krill recruitment) of each scenario and across all four parameterizations noted above (and described in Watters et al. [39]; i.e., we ran 4004 Monte Carlo trials per scenario). As with previous implementation of KPFM2, we averaged results across trials and parameterizations for each scenario to account for model uncertainty (see [39, 41]). A schematic of this process is provided in Fig 2.

We computed outcomes in terms of predator abundance and catches taken by the fishery at two points in time to assess outcomes after 30 years and at the end of the run. We also considered results aggregated across the entire model spatial arena, for broad overall changes, and at the SSMU scale, for spatial differences. To make outcomes comparable across the differing management strategies, we report results relative to the respective No FBM or No MPA base case scenario (e.g., the ratio of catch under FBM to catch in the No FBM reference; a 'counterfactual' as described in [35], Fig 2). Using this approach, results equal 1.0 if the management

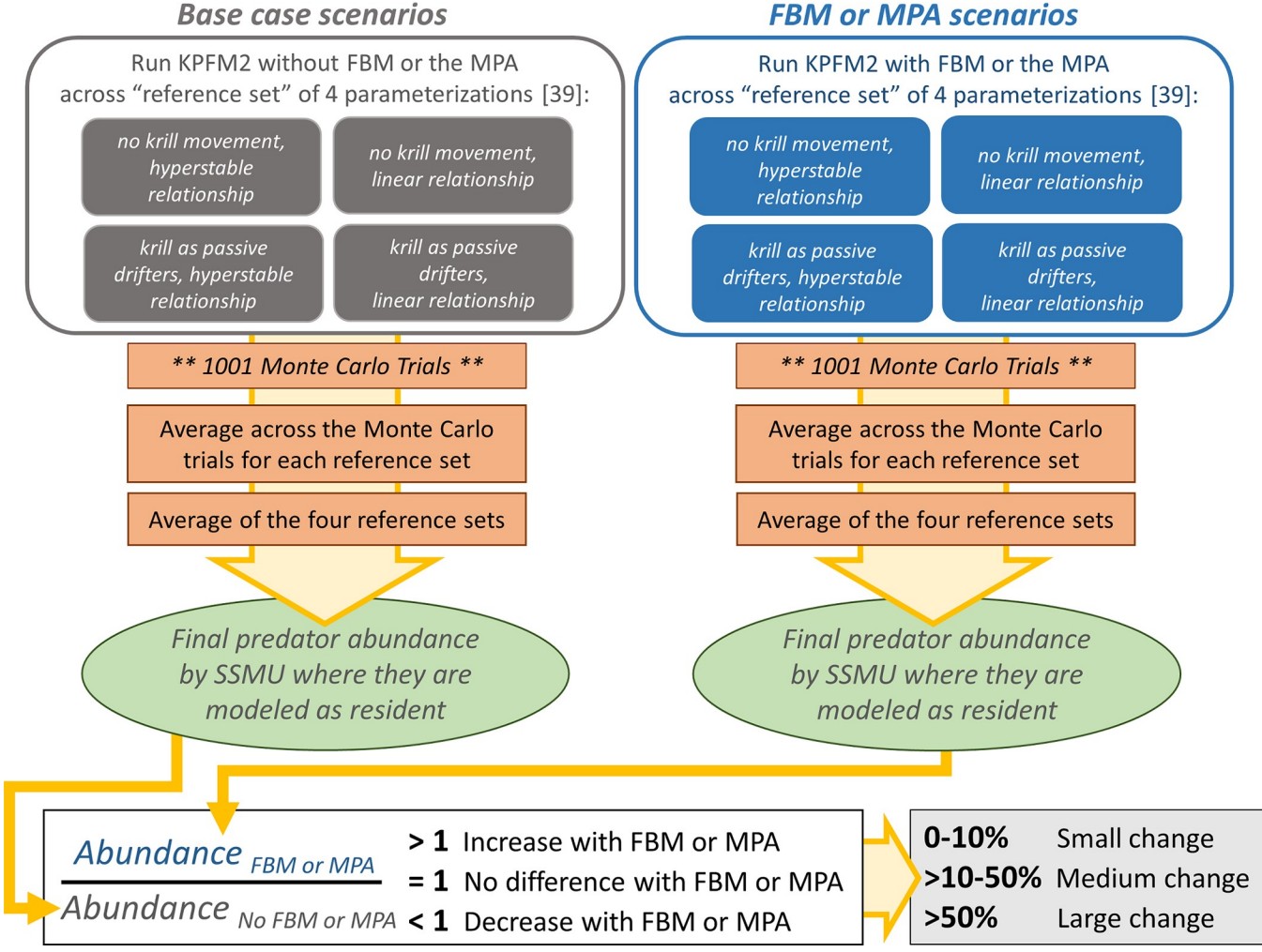

**Fig 2. Schematic of the modelling process.** For both the 'base case' (left column, top grey boxes) and FBM or MPA (right column, top blue boxes) scenarios, the model is run across all four parameterizations (top boxes) and across 1001 Monte Carlo trials. These are then averaged for final results (green circle), and the relative change assessed (bottom row of boxes).

action, FBM or the MPA, did not affect outcomes, while results >1.0 and <1.0 respectively indicate increases (positive outcomes) and decreases (negative outcomes). As an additional metric of fishery performance, we also computed the SSMU-specific probabilities that the fishery would suspend operations because krill density fell below 15 g·m$^{-2}$, a level previously identified as an important threshold for the fishery [51]. The risks of incurring such "threshold violations" indicate whether redistributing catches may increase the costs of fishing beyond those related to changing catches themselves.

To aid in comparing results, we also determined an arbitrary scale: an absolute change of 0.01 to 0.10 denotes a "small" change (either increase or decrease), > 0.10 to 0.50 a "medium" change, and > 0.50 a "large" change. We stress that this scale is simply for comparison, and we do not attribute significance to these ranges. The significance of a change in abundance will depend on the species, and that of a change for the fishery will be interpreted differently by different people. Further, we note that our results are model projections and should be taken as strategic advice, meaning readers should consider overall patterns not specific numbers.

## Results

Relative abundances of krill predators were sensitive to the management strategies investigated here. For results aggregated across the model arena, FBM-Krill caused little change in the total abundances of penguins or seals relative to the No FBM scenario ("small" or no absolute changes, all < ±0.10%), but more obvious declines were projected for both species groups with FBM-Pengs (Fig 3A), with a small decline (-0.04) in penguins and a medium decline (-0.11) in

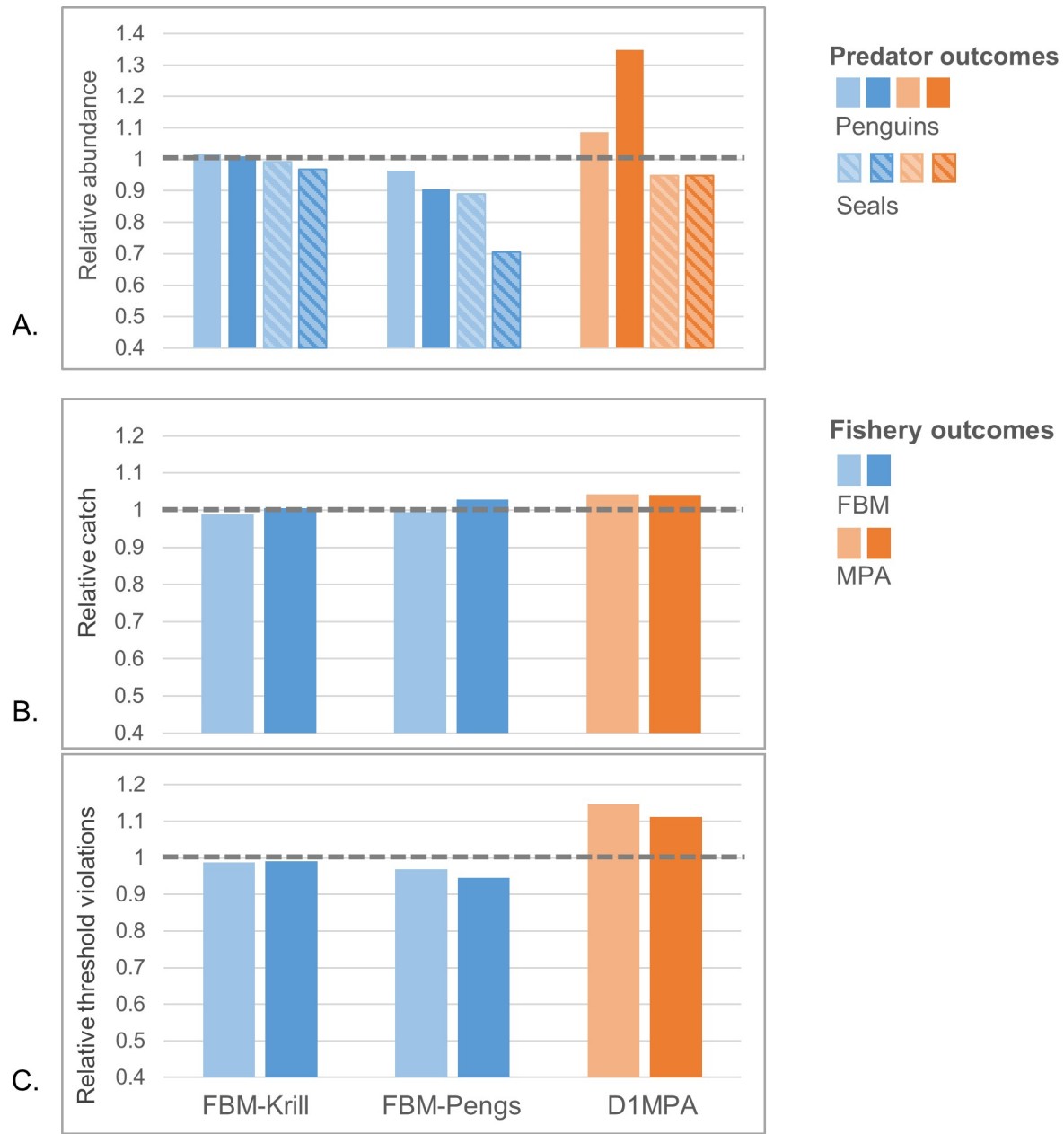

**Fig 3.** Relative total change in predator abundance (A) and fishery performance (B and C) across modeled scenarios and aggregated at the scale of the full model arena. Lighter shades are at 30 years in the model run, and darker shades at 100 years. Fishery performance is measured both as relative catch (B) and the probability of a threshold violation (C). Feedback strategies are indicated in blue, and the MPA in orange. All results are relative to the No FBM or No MPA scenarios, with the dashed grey line at 1.0 indicating no impact of FBM or the MPA.

seals. Projected outcomes were similar after 30 and 100 years for FBM-Krill, but total declines under FBM-Pengs became greater over time (-0.09 for penguins and -0.30 for seals). In contrast, the MPA was projected to provide small increases in penguin abundance relative to the base case scenario without an MPA (+0.09), but with medium increases (+0.35) by the end of the model century. The relative abundance of seals declined with the MPA, but this decrease was small (-0.05) and did not change over time in the model runs. Whales saw small increases under FBM-Pengs (+0.02), with larger but still small increases under the MPA (+0.05), and both increases were greater over time (+0.04 and +0.09). No changes were found for whales with FBM-Krill or for the fish group under any of the scenarios (all absolute changes < 0.01) (S1 Fig).

In general, we projected only small changes in fishery performance. For the krill fishery, projected catches over the whole model arena were relatively unaffected by the two FBM strategies and the MPA (Fig 3B). FBM-Krill resulted in small or no change (-0.01 and +0.005 for FBM-Krill and -0.01 and +0.03 for FBM-Pengs, for 30 and 100 years, respectively), and the MPA only a small increase that did not change over time (+0.04). We found comparatively greater differences between management strategies in terms of the probabilities of threshold violations, with small changes under FBM-Krill (-0.01 across time in the model), slightly larger but still small decrease with FBM-Pengs (-0.03 at 30 years and -0.05 at the end of the model run), and medium increases with the MPA (+0.15 and +0.11) (Fig 3C).

At the smaller SSMU spatial scale, our model projected the relative abundances of krill predators to vary spatially and across scenarios (due to the large number of results, we have projected outcomes as maps, with results available in the S1 Data). FBM-Krill generally provided slight benefits to penguins and seals in several SSMUs (Fig 4A–4D), but outcomes from FBM-Pengs were more mixed, with relative abundances increasing in some SSMUs and decreasing in others (Fig 5A–5D). While there were only slight difference in SSMU-specific outcomes from FBM-Krill after 30 and 100 years (Fig 4A–4D), we projected that FBM-Pengs intensified changes in predator abundance over time (Fig 5A–5D). With the MPA, variability in the relative abundances of penguins and seals among SSMUs was intermediate to the levels of variability under FBM-Krill and FBM-Pengs, but with greater positive outcomes projected across most SSMUs for penguins (Fig 6A–6D). These results were similar for whales and fish (slight increases in some areas with FBM-Krill, variable outcomes in FBM-Pengs and the MPA), and the SSMU-specific outcomes for these groups generally changed little over time (S2–S4 Figs).

We projected performance of the krill fishery would also vary by SSMU. FBM-Krill resulted in various changes in relative catch, with decreases in most coastal SSMUs in Subarea 48.1 and 48.2, and increases elsewhere (Fig 4E–4F). FBM-Pengs projected even more pronounced changes in relative catch, but, again, decreases were projected in some coastal SSMUS and increases in others (Fig 5E and 5F). We found comparatively smaller changes in relative catch with the MPA, with catches increasing in many SSMUs and declines in only one (Fig 6E and 6F).

We also explored whether changes in relative abundance of krill-dependent predators were related to changes in relative catch. Fig 7 compares relative change in catch with either FBM or the MPA against relative change in penguin (Fig 7A and 7B) or seal (Fig 7C and 7D) abundance, with points to the right of the dashed vertical x = 1 line denoting greater catch with FBM or the MPA, and points above the horizontal y = 1 indicating increases in penguin or seal abundance. For penguins, several patterns emerge (Fig 7A and 7B). First, the majority of the blue points (either squares, FBM-Krill, or triangles, FBM-Pengs) are left of the x = 1 line, indicating relatively lower catches with FBM in these SSMUs, while the orange circles denoting the MPA are almost all to the right of this line, indicating relatively greater catches with the MPA.

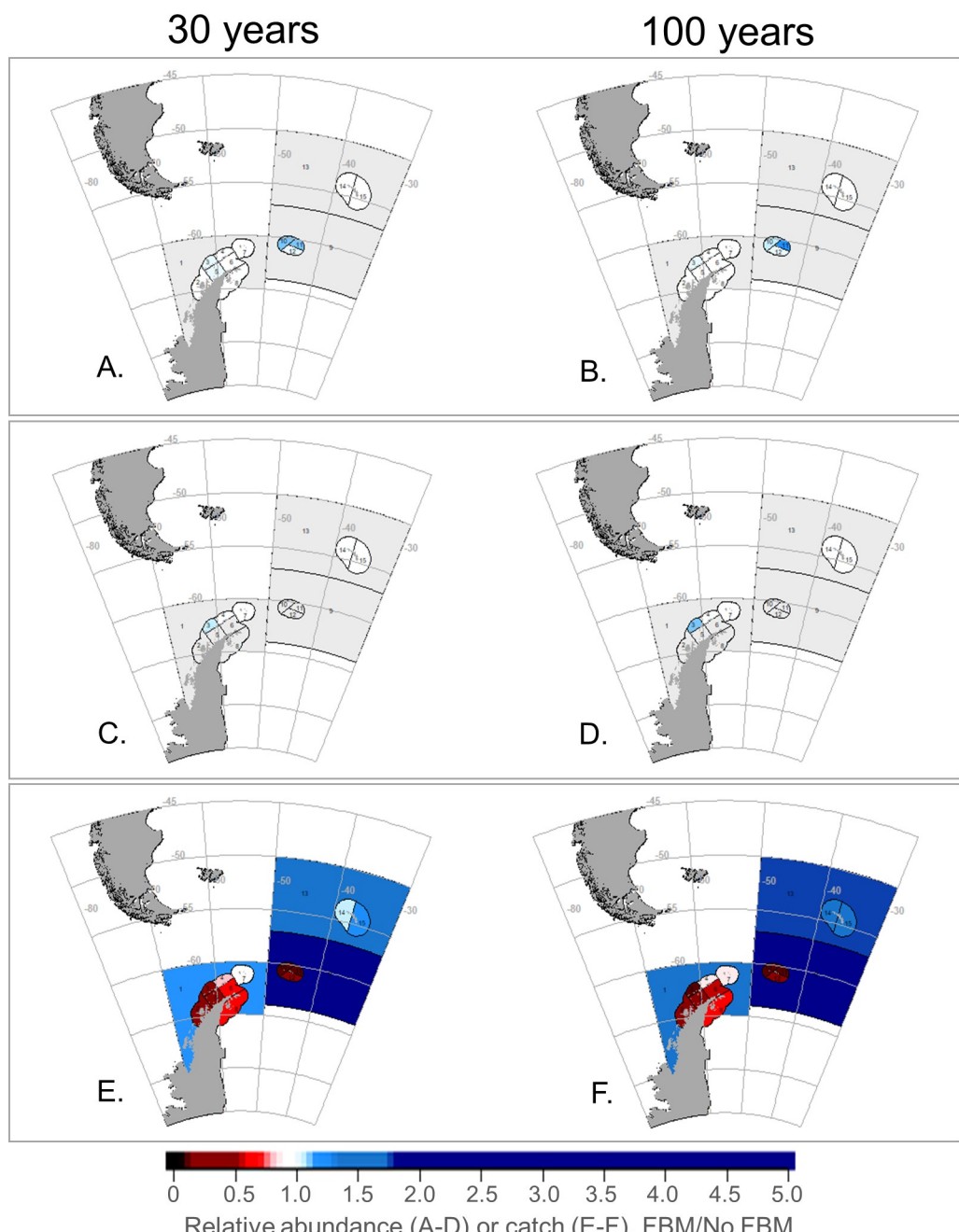

**Fig 4. SSMU-specific outcomes of FBM-Krill for predators and the fishery under a modeled climate change impact.**
Projected penguin (A, B) and seal (C, D) abundances, and krill catches (E, F) given climate-change impacts on krill growth, with outcomes at 30 years in to the model run in the left column (A, C, E) and at 100 years in the right (B, D, F). Blues represent increases relative to the No FBM base case scenario and reds decreases; white and light colors indicate no or little change. Light grey denotes areas where the species group is not modeled as resident. Note changes are relative to the No FBM base case within each SSMU, not over the entire model arena.

That is, both FBM-Krill and FBM-Pengs result in more SSMUs with less catch, whereas more SSMUs yield relatively greater catches with the MPA. Second, all SSMUs under either FBM strategy where penguins increased in the model (blue squares and triangles above the y = 1 line) were also in SSMUs with lower catches. In contrast, both catches and penguin

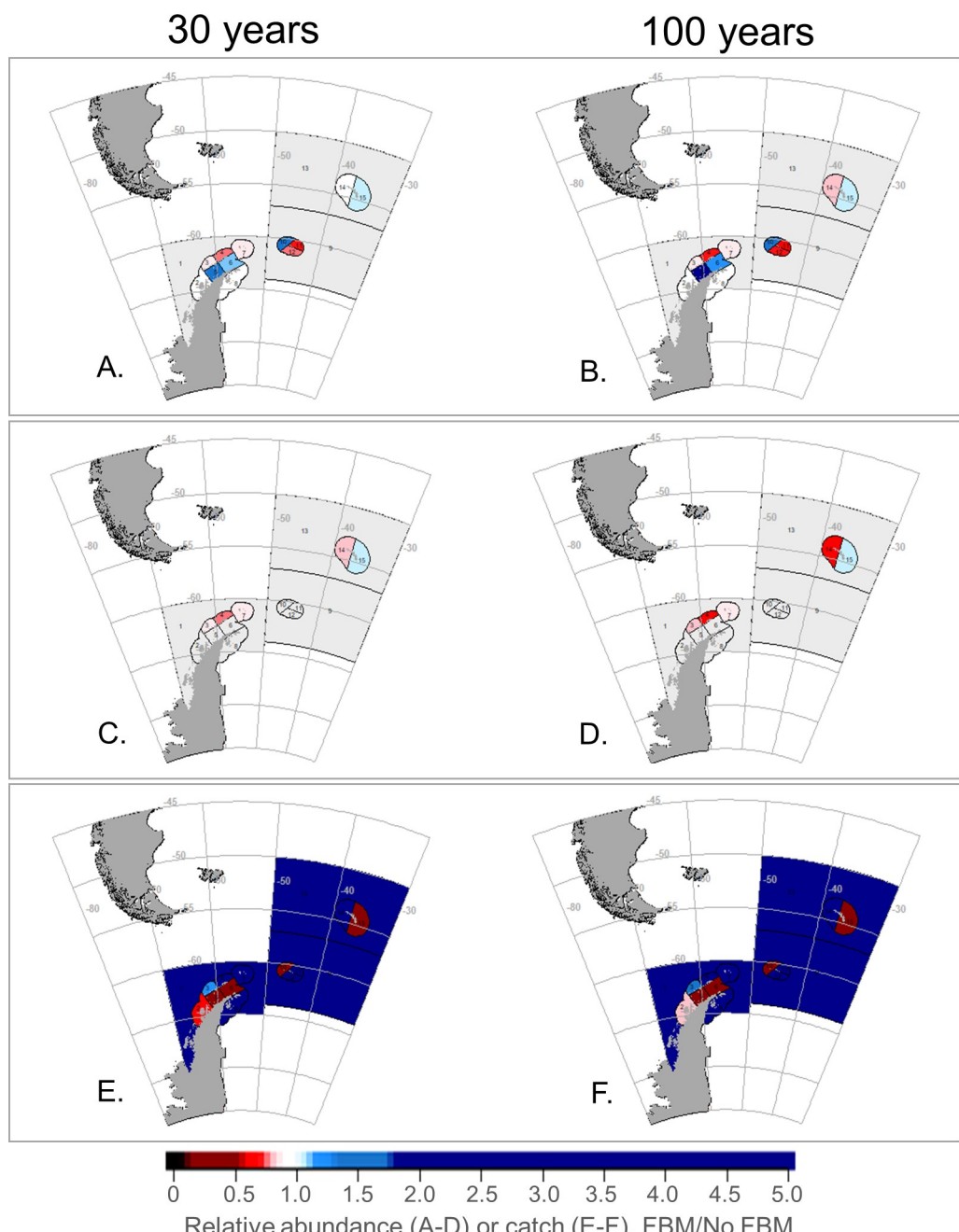

**Fig 5. SSMU-specific outcomes of FBM-Pengs for predators and the fishery under a modeled climate change impact.** Projected penguin (A, B) and seal (C, D) abundances, and krill catches (E, F) given climate-change impacts on krill growth, with outcomes at 30 years in the left column (A, C, E), and 100 years in the right (B, D, F). All other details as in Fig 4.

abundances increased in most SSMUs with the MPA (orange circles are to the right of the x = 1 line and above the y = 1 line). These patterns somewhat held for seals, except that the MPA did not increase seal abundance (i.e. orange circle are generally not above the y = 1 line in Fig 7C and 7D), a result also noted in [38]. Finally, a third pattern emerged for penguins

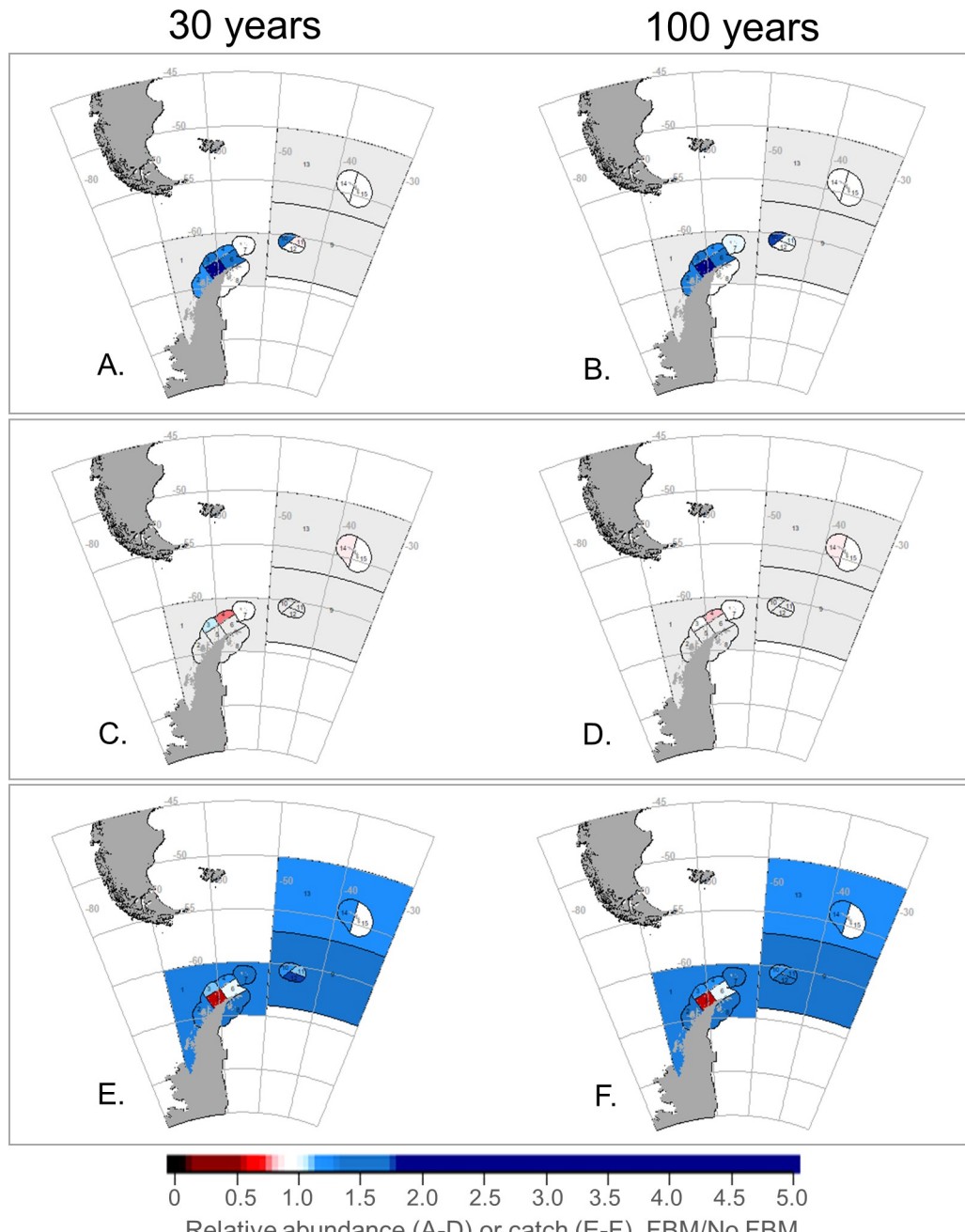

**Fig 6. SSMU-specific outcomes of the MPA for predators and the fishery under a modeled climate change impact.**
Projected penguin (A, B) and seal (C, D) abundances, and krill catches (E, F) given climate-change impacts on krill
growth, with outcomes at 30 years in to the model run in the left column (A, C, E), and at 100 years in the right (B, D, F).
All other details as in Fig 4, aside from the base case being the No MPA scenario.

that was not apparent for seals: in SSMUs where penguins did increase and catches were low-
est, FBM-Pengs did project greater increases in penguins than FBM-Krill. These patterns were
conserved over the course of the model run, with slight increases in abundance for penguins
(outcomes for whales and fish in S5 Fig).

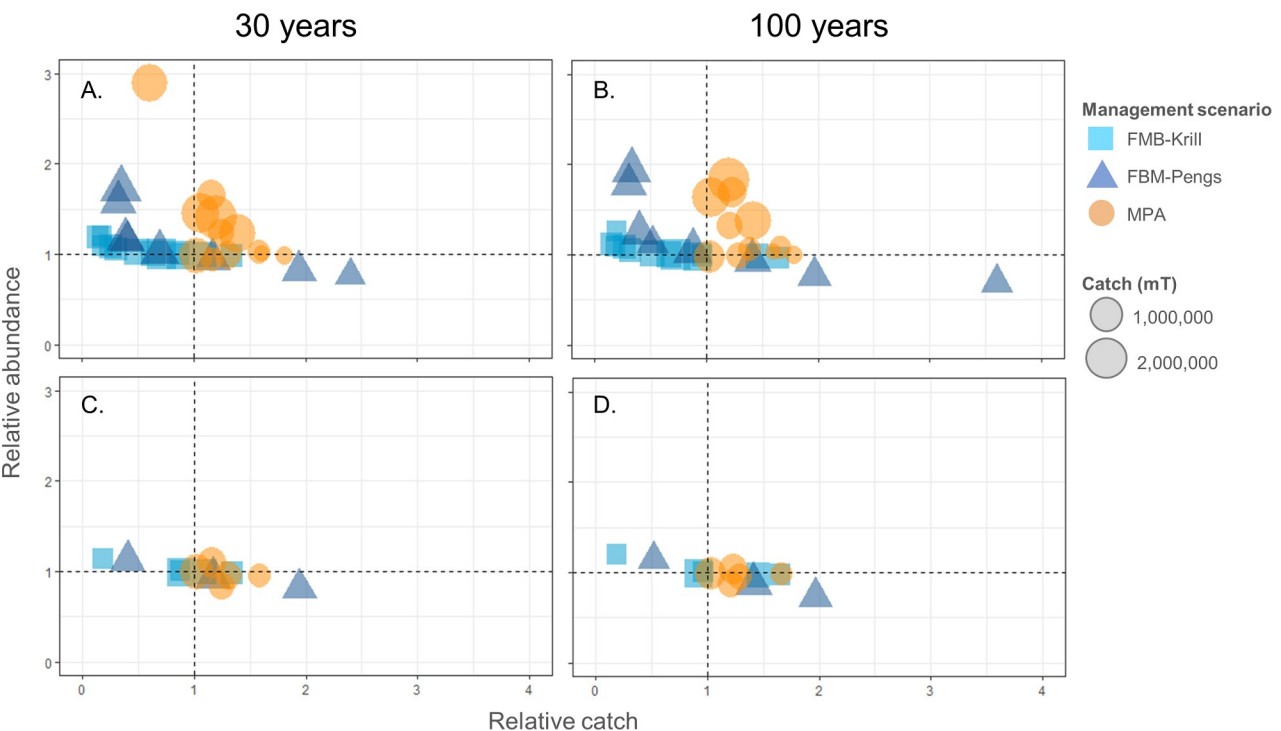

**Fig 7. Relationship between relative catches and the relative abundances of penguins and seals given the two FBM strategies and an MPA and a modeled impact of climate change.** Relative catches (FBM/No FBM or MPA/No MPA, x-axis) and relative changes in the abundances (FBM/No FBM or MPA/No MPA, y-axis) of penguins (A, B) and seals (C, D) given FBM-Krill (light blue squares), FBM-Pengs (dark blue triangles), and the MPA (orange circles) at 30 years (left column, A and C) and at 100 years (right column, B and D). The dashed lines represent no change with FBM or the MPA in catch or abundance at x = 1 and y = 1, respectively.

## Discussion

The management strategies considered here, if implemented, seem likely to land decision makers at different locations within the multivariate tradeoff space inherent in managing the krill fishery. We found that adjusting to future change by monitoring krill density (FBM-Krill) offered slight benefits to some predator populations while simultaneously maintaining catches taken by the fishery and mitigating the risk that the fishery would be displaced to areas of low krill density. In contrast, when adjustments to future change were based on the abundances of breeding penguins (FBM-Pengs), we projected decreases in the relative abundances of some penguin populations and substantially more spatial variability in the relative abundances of all krill predators alongside only minor increases in relative catches and a lower probability of threshold violations, i.e., fishing in areas of low krill density. When we simulated an MPA scenario, almost all penguin populations and catches across most SSMUs increased, but we also projected declines in the relative abundance of seals and an increased probability of threshold violations.

The tradeoffs described above have taught us at least three useful and generalizable lessons. First, it seems easy to design a well-intentioned but undesirable FBM strategy. We believe that FBM-Pengs represents one such strategy. Although FBM-Pengs was designed to adjust to future change by observing penguins, this approach was the worst at conserving krill predators, including penguins, of the three management strategies we considered. We discuss why this might be the case later in our Discussion. Second, it seems that FBM strategies which are explicit about prey but not their predators may nevertheless be useful for conserving predators

while being adjustable in a changing future. While it did not deliver large benefits in the model, we believe FBM-Krill still demonstrates this potential. FBM-Krill did not exacerbate declines in the relative abundances of krill-dependent predators and some populations increased, even with the fishery fully developed, identifying a potential strategy for adjusting to change while maintaining predator populations and the fishery. Our final lesson is that management strategies designed to be robust to future change, like the MPA considered here, may provide ecological benefits, but may also involve strong tradeoffs between resource conservation and utilization.

Alongside previous work showing that fishing for krill can increase risks to predators (*e.g.*, [41, 42]), our results continue to indicate that benefits for predators may accrue in locations where forage-fish catches are reduced. Both FBM scenarios considered here, but more strongly FBM-Pengs, showed increases in the relative abundances of predators in SSMUs where relative catches decreased (Fig 7). A different pattern emerged for the MPA. The SSMUs with increased predator abundances, particularly of penguins, were also projected to support comparatively larger catches of krill. We suggest that this result arises from how displaced catches were spatially reallocated in the model. Catches could still be taken from SSMUs in which fishing recently occurred but which were not fully encompassed by the MPA. Therefore, displacement was minimal relative to the stricter redistribution in the FBM-Pengs scenario. Collectively, our results demonstrate an underlying difference in how FBM and MPA strategies may provide ecological benefits: by shifting the location of fishing due to changes in an indicator and possibly to completely new areas (FBM), or away from critical areas altogether but allowing for effort to continue nearby (MPA).

As noted previously, our findings also indicate that shifting the spatial distribution of catches with FBM strategies may not always reduce ecological risk as intended. The FBM-Pengs scenario failed to provide broad benefits for krill predators–even for the indicator species, penguins–and, instead, caused some populations to decline. We believe this result is due to our implementation of FBM-Pengs. We conserved the overall catch limit–that is, the fishery harvested the same amount of krill—and spatially allocated catch limits based on SSMU-specific changes in penguin abundance relative to overall changes in penguin abundance throughout the model arena. Those SSMUs with the greatest declines in penguin abundance saw the greatest reductions in catch, but catch allocations to SSMUs with relatively smaller penguin declines may not have decreased and may even have increased if there were larger penguin declines elsewhere. That is, our modeled FBM strategy successfully displaced catches, but did not always shift catches away from vulnerable populations. In some cases, catches were displaced into SSMUs where the abundance of predators was indeed declining.

We do note that the SSMUs with the greatest reductions in catch did see strong positive outcomes for many krill-dependent predators, particularly penguins, indicating that FBM-Pengs achieved ecological goals for some populations in the model (Fig 7). Given this, an alternative formulation of FBM-Pengs might be to reduce SSMU-specific catch limits based on changes in penguin abundance in that SSMU regardless of such changes in other areas. This alternative would ensure decreased catches with decreases in penguin abundance—but might also reduce the overall catch when penguin declines are widespread. We conserved the overall catch limit because we assumed such an approach would be more desirable to some CCAMLR Members and the fishing industry. However, if region-wide declines in penguin abundance continue as a function of climate change, increasing cetacean populations, and other drivers (*e.g.*, [7,59–62]), our results imply that overall reductions in catches may be required by an FBM strategy that aims to mitigate declines in predator abundance (e.g., as discussed for CCAMLR in [9]). More broadly, the outcomes from FBM-Pengs demonstrate the significance of carefully considering the indicators and decision rules used within FBM strategies; decision

rules that are explicit about predators will not necessarily ensure that the objectives of ecosystem-based fisheries management are achieved.

It is tempting to believe the routine application of a preset decision rule (*e.g.*, [21]) is beneficial. In our experience, preset decision rules can efficiently increase transparency and the use of "best available science" in fisheries management. However, our results show that when preset decision rules are poorly specified, as with FBM-Pengs, negative consequences can be exacerbated over the course of time (e.g., Fig 5). Management strategies that fail to conserve predator populations in the near term seem likely fail over the long term as well, even if these strategies are designed to adjust to future change. Yet our projections with FBM-Krill and the MPA also indicate that management strategies which are successful in the near term may help to mitigate negative outcomes in the long run as well (Figs 3, 4, 6 and 7).

Our work also indicates that mandated shifts in the spatial distribution of fishing required by an FBM strategy or an MPA may be burdensome to the fishing industry, and redistributing fishing effort can come with significant costs [63–65]. All the scenarios we explored here would displace fishing from the coastal SSMUs surrounding the Antarctic Peninsula and islands in Subareas 48.1, 48.2, and 48.3. Watters et al. [39] demonstrated redistributing fishing farther offshore, to "pelagic SSMUs", could increase the ability of CCAMLR to achieve its conservation objectives, but these offshore areas have lower krill densities than coastal areas. Consequently, all three of the scenarios we explored here would, if implemented, be likely to shift fishing offshore but also decrease performance of the fishery. In addition, the redistribution of effort necessitated by an FBM strategy, including that to pelagic areas with lower krill densities, would be required and change after each assessment. In contrast, an MPA would close areas outright, and, after an MPA is established, the redistribution of fishing activity would likely be determined by the fleet itself. Indeed, previous results suggest that an effective MPA could allow fishing vessels to self-select grounds in open areas, without a need to further distribute catches spatially [38]. Thus, a primary difference between the FBM and MPA scenarios we considered is the degree of prescription about where fishing may occur. Such differences, as well as the spatial expression of possible costs, are critical for decision makers to consider in light of stakeholder needs and preferences.

Of course, reality is more complicated than our model is able to reflect, and modeling of such complex systems comes with caveats and assumptions. The caveats associated with KPFM2 are well defined in the literature [38, 39, 41, 43, 44]. These caveats include using aggregated predator groups and specifying certain functional relationships. Using aggregate groups may mean we are missing important dynamics for certain species within those groups, and alternative functional relationships may be more appropriate in some cases or in the future. We also make assumptions about krill densities and the spatial distribution of krill, as well as fishing patterns and the redistribution of fishing effort displaced by the MPA, the latter of which also has implications for fishery performance [38]. Moreover, for simplicity, we simulated climate-change effects of krill GGP using only RCP 8.5 [48], which assumes no action is taken on climate change. Therefore, on one hand, the results here could be seen as a "worst case" scenario. However, on the other hand, we implemented only one potential consequence of climate change on krill alone. In reality the effects of climate change will be more complicated than we have simulated. These effects may make populations more or less vulnerable, exacerbating or mitigating the outcomes projected here (as discussed in [41]). Despite all the caveats of our work, we assert that the broad implications of our results remain useful.

There are also a plethora of potential FBM and MPA scenarios that could be modeled. Here, both were based on community feedback and interest (FBM), and proposals based on community engagement (the D1MPA and the South Orkney Islands Southern Shelf MPA, [55, 56, 58]). Our previous research showed the D1MPA could be improved to better achieve

various ecological outcomes [38]. Improved FBM scenarios are equally possible, and the challenges with FBM found here might be overcome by identifying other strategies that more readily provide region-wide benefits. Our goal was not to find a "best" scenario, but rather to compare possible outcomes and thereby supply strategic guidance regarding FBM strategies and MPAs given a changing climate. We contend the strategies considered here provide insight that is useful regardless of whether there are better or more preferable alternatives.

Our work evokes an interest in contrasting management strategies that dynamically adjust to change (e.g., FBM) against those that are statically robust to change (e.g., MPAs). Neither type of strategy is a panacea, and we have identified a potentially pathological FBM strategy (FBM-Pengs), while other research demonstrates concerns with MPAs (*e.g.*, [63, 66]). Nevertheless, both FBM and MPAs may offer significant benefits, and we do not think contrasts between dynamic and static management strategies can easily be generalized. Some FBM strategies may require fewer data and analyses than more traditional approaches [17]; limit "haggling" that slows the management process [21]; are more tenable to the fishing industry due to the continuity of the approach [19]; stabilize harvested systems and avoid fishery collapse [22, 26]; and meet multiple management objectives [18]. We reason all of these benefits can also be the outcomes of a well-designed MPA. Conversely, we contend that the benefits of an MPA can be the outcomes of a well-designed FBM strategy, including reducing or reversing adverse human impacts [27, 67–69], buffering against uncertainty [24, 25], providing ecological benefits [70–72], and improving fishery yields [73–75].

We emphasize that FBM strategies and MPAs necessitate continued engagement in the management process once an approach is implemented, but the nature of that engagement warrants consideration. By definition, feedback strategies require monitoring of indicators; an MPA obliges monitoring to assess the effectiveness of the protected area. That is, monitoring to support an FBM strategy is, at a minimum, determined by what is needed to implement the decision rule, whereas monitoring to support an MPA may be more flexible and left to the discretion of the research community to determine whether the protected area is successful. These requirements should also be considered when making decisions.

Ultimately, the implementation of either an FBM strategy or an MPA will depend on the values of stakeholders involved as well as practical and political realities. Work such as ours aims to facilitate discussion by providing insight on potential benefits and costs of alternative management strategies, supporting CCAMLR's existing management frameworks that endeavors to establish FBM and/or MPAs, and exemplify actionable science for policy [76]. This is particularly important as either strategy will require time and resources to implement and maintain, and CCAMLR is currently working to harmonize the D1MPA with FBM because the proposed MPA encompasses an important krill fishing area. Other researchers have also discussed the value of combining approaches, suggesting the use of either FBM [26] or adaptive management [77] to improve MPAs. One possibility might be to implement the D1MPA and later implement an FBM or more fully adaptive strategy that aims to adjust the MPA over time. In fact, an early version of the D1MPA proposal before CCAMLR [78] takes a step towards adaptive management. That proposal includes sequentially closing areas to assess the utility of an MPA and incorporating special zones in which experimental krill fishing could be conducted with the purpose of parameterizing one or more decision rules for an FBM strategy.

Strategic advice is crucial for decision makers looking to advance ecosystem-based management, including approaches such as FBM and MPAs, and will likely prove useful in a changing, uncertain future. Advice that compares potential strategies allows for more informed decision making and advances towards fully realized ecosystem-based adaptive management, and our work supports colleagues who have asserted that ecosystem models are valuable for providing

such advice [34–36] and current practices in CCAMLR [10, 76]. Stakeholders in the Southern Ocean vary on what features of FBM and MPAs are most appealing and acceptable [79], but, critically, CCAMLR's participatory process allows for broad consideration of possible options and their outcomes. This process also provides a framework to collectively determine the best way forward.

## Supporting information

**S1 Fig. Relative changes in the abundance of additional predator groups across modeled scenarios and aggregated at the scale of the full model arena.** Lighter shades are at 30 years in the model run, and darker shades at 100 years. Feedback strategies are indicated in blue, and the MPA in orange. All results are referenced to the No FBM or No MPA scenarios, with the dashed grey line at 1.0 indicating no impact of FBM or the MPA.
(TIF)

**S2 Fig. SSMU-specific outcomes of FBM-Krill for additional predator groups under a modeled climate change impact.** Projected whale (A, B) and fish (C, D) abundances given climate-change impacts on krill growth, with outcomes at 30 years in to the model run in the left column (A, C) and at 100 years in the right (B, D). Blues represent increases relative to the No FBM base case scenario and reds decreases; white and light colors indicate no or little change. Light grey denotes areas where the species group is not modeled. Note changes are relative to the No FBM base case within each SSMU, not the entire model arena.
(TIF)

**S3 Fig. SSMU-specific outcomes of FBM-Pengs for additional predator groups under a modeled climate change impact.** Projected whale (A, B) and fish (C, D) abundances given climate-change impacts on krill growth, with outcomes at 30 years in to the model run in the left column (A, C), and at 100 years in the right (B, D). All other details as in S2 Fig.
(TIF)

**S4 Fig. SSMU-specific outcomes of the MPA for additional predator groups under a modeled climate change impact.** Projected whale (A, B) and fish (C, D) abundances given climate-change impacts on krill growth, with outcomes at 30 years in to the model run in the left column (A, C), and at 100 years in the right (B, D). All other details as in S2 Fig, aside from the base case being the No MPA scenario.
(TIF)

**S5 Fig. Relationship between relative catches and the relative abundances of whales and fish given the two FBM strategies and an MPA and a modeled impact of climate change.** Relative catches (FBM/No FBM or MPA/No MPA, x-axis) and relative changes in the abundances (FBM/No FBM or MPA/No MPA, y-axis) of whales (A, B) and fish (C, D) given FBM-Krill (light blue squares), FBM-Pengs (dark blue triangles), and the MPA (orange circle) at 30 years (left column, A and C) and at 100 years (right column, B and D). The dashed lines represent no change in catch or abundance at x = 1 and y = 1, respectively.
(TIF)

**S1 File.**
(DOCX)

**S1 Data.**
(XLSX)

## Acknowledgments

We thank M. Santos (Instituto Antártico Argentino), A. Capurro (Dirección Nacional del Antártico, Argentina), and C. Cárdenas (Instituto Antártico Chileno) for graciously granting permission to cite working papers and shapefiles that describe the D1MPA. We also thank A. Dahood for helping to create Fig 1.

## Author Contributions

**Conceptualization:** Emily S. Klein, George M. Watters.

**Data curation:** Emily S. Klein, George M. Watters.

**Formal analysis:** Emily S. Klein.

**Funding acquisition:** Emily S. Klein, George M. Watters.

**Investigation:** Emily S. Klein, George M. Watters.

**Methodology:** George M. Watters.

**Project administration:** George M. Watters.

**Resources:** George M. Watters.

**Software:** Emily S. Klein, George M. Watters.

**Supervision:** George M. Watters.

**Validation:** Emily S. Klein.

**Visualization:** Emily S. Klein, George M. Watters.

**Writing – original draft:** Emily S. Klein.

**Writing – review & editing:** Emily S. Klein, George M. Watters.

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
