## [Decision Letter · Decision Letter 0]

12 Sep 2019

PONE-D-19-22435

Comparing feedback and spatial approaches to advance ecosystem-based fisheries management in a changing Antarctic

PLOS ONE

Dear Dr Klein,

Thank you for submitting your manuscript to PLOS ONE. I have now received reports from two referees who found your work interesting and worthy of publication. They both noted a number of issues that need to be addressed before such a stage can be reached. Referee 2 especially noted the need for more details in the Methods and the construction of your model, especially regarding how krill predator dynamics are modelled (how do you deal with the different taxonomic groups of predators). Therefore, I selected the Major Revision decision and invite you to submit a revised version of the manuscript that addresses all the points raised during the review process. Please note that comments by reviewer 2 are included in the pdf file enclosed to this message.

We would appreciate receiving your revised manuscript by Oct 27 2019 11:59PM. To enhance the reproducibility of your results, we recommend that if applicable you deposit your laboratory protocols in protocols.io, where a protocol can be assigned its own identifier (DOI) such that it can be cited independently in the future. For instructions see: http://journals.plos.org/plosone/s/submission-guidelines#loc-laboratory-protocols

We look forward to receiving your revised manuscript.

Kind regards,

Yan Ropert-Coudert, PhD

Academic Editor

PLOS ONE

Journal Requirements:

 [EK was supported by funding from the Pew Charitable Trusts, contract ID #31740. This funder had no role in study design, data collection and analysis, or preparation of the manuscript. Publication under peer review was a requirement of this funding source, but the funder did not take part in deciding where this manuscript would be submitted or any part of the submission process.]. 

3. We note that Figures [1, 3, 4, 5 and S2, S3, and S4] in your submission contain map/satellite images which may be copyrighted. All PLOS content is published under the Creative Commons Attribution License (CC BY 4.0), which means that the manuscript, images, and Supporting Information files will be freely available online, and any third party is permitted to access, download, copy, distribute, and use these materials in any way, even commercially, with proper attribution. For these reasons, we cannot publish previously copyrighted maps or satellite images created using proprietary data, such as Google software (Google Maps, Street View, and Earth). For more information, see our copyright guidelines: http://journals.plos.org/plosone/s/licenses-and-copyright.

You may seek permission from the original copyright holder of Figures [1, 3, 4, 5 and S2, S3, and S4] to publish the content specifically under the CC BY 4.0 license. 

If you are unable to obtain permission from the original copyright holder to publish these figures under the CC BY 4.0 license or if the copyright holder’s requirements are incompatible with the CC BY 4.0 license, please either i) remove the figure or ii) supply a replacement figure that complies with the CC BY 4.0 license. Please check copyright information on all replacement figures and update the figure caption with source information. If applicable, please specify in the figure caption text when a figure is similar but not identical to the original image and is therefore for illustrative purposes only.

Reviewers' comments:

Reviewer's Responses to Questions

**Comments to the Author**

1. Is the manuscript technically sound, and do the data support the conclusions?

Reviewer #1: Yes

Reviewer #2: Partly

2. Has the statistical analysis been performed appropriately and rigorously? 

Reviewer #1: N/A

Reviewer #2: No

3. Have the authors made all data underlying the findings in their manuscript fully available?

Reviewer #1: No

Reviewer #2: Yes

4. Is the manuscript presented in an intelligible fashion and written in standard English?

Reviewer #1: Yes

Reviewer #2: Yes

5. Review Comments to the Author

Reviewer #1: GENERAL COMMENTS

Before reading the whole article, I kind of felt after reading the abstract that the FBM strategies (FBM-Krill) provide greater benefits to predators than the MPA strategy (lines 44 – 48), yet after reading the article I know that’s not the case (this should be made clearer). In fact, the results indicate that MPAs seem to be a better strategy for both predators (except seals) and fisheries – they maintain (slightly increase) catch and increase predator abundance. Of course, it’s not as black and white as this (especially when considering increased fishing in areas of low krill density), but the tone of the article (abstract/results/discuss) comes across as slightly ‘pro-FBM’ and slightly downplaying MPAs, and the authors should consider whether the article needs a slight reframing to be more balanced.

This is particularly important where the overall benefits to penguin abundance under MPAs are the only substantial benefits from any of the three strategies to either biodiversity or fisheries and it should be stated more clearly. Particularly when you determine that total catch was only maintained by the FBM strategies and only slightly increased with MPAs – which starts to raise the question about whether fisheries and fisheries managers would really be interested in applying such strategies if there are also no obvious substantial outcomes for predators (such as the two FBM strats)? It costs time and resources to apply a strategy, so the outcomes/benefits need to be tangible and substantial for industry or CCAMLR to take note (thus it is important to highlight the benefits provided under any strategy).

What kind of seals are you talking about – are they ice-dependant or ice independent? And are you including all 4 Antarctic penguin species?

It would probably be helpful having a flow chart detailing the important steps/variables in the methods (e.g. number of simulations, main indicators, names of the scenarios, timeframe, what the output variable being measured is etc). This would help the reader follow the process.

You didn’t undertake any statistical analysis to determine how much each of the strategies differed from the baseline (No FBM/No MPA), and instead use terms like ‘minimal increases’ etc, which is fine – though it would be useful to provide a scale for the reader to determine what a substantial impact on predator abundance/ fisheries catch is. For example, Fig 2 shows a ~0.35 increase in penguin abundance under the MPA scenario (for 2100) – is this equivalent to a 35% increase in abundance? Which certainly appears to be a substantial increase. If a 35% increase is the wrong interpretation, the scale needs to be clarified as this is how it will be interpreted. You could also consider undertaking statistical analyses to compare the strategies to the baselines (ANOVAs would also allow you to compare between scenarios).

Data availability – it might be helpful for managers and conservationists to have access to the spatial data layers of the SSMU outcomes so that they can utilise them in spatial planning or future studies

SPECIFIC COMMENTS

ABSTRACT

Line 46 – make it ‘outcomes of our second FBM option’ to make it clearer that you are referring to the FBM option rather than the second overall options that you mention in line 40. Maybe reword the line 40 one as ‘strategies’ to keep it consistent with the rest of the article?

INTRODUCTION

Lines 61-65 – Some of these statements could use references

68 – might be useful to give an example of observed climate impacts in the Southern Ocean rather than just referring to refs

95-96 – its not really the decision makers that are robust/adapt to the change, it’s the target/ecosystem that needs to be robust or adapt

97-99 – needs referencing, and would be better split into two sentences.

101-102 – where are the FBM specific objectives/priorities stated by CCAMLR?

104 – what important krill fishing area?

109-112 – beginning with ‘these models’ – split this sentence into 2 – 3 sentences to make it easier to read

117 – should specify that krill density and penguin abundance are the indicators.

118 – how can you project a new MPA based on previously shown ecological benefits (the ecological benefits of a proposed MPA haven’t been established yet)? This needs more explaining/ simplifying to make it less confusing

123 – this is a very broad statement?

Fig1. There is no context to this figure in the intro, it should probably be first referenced in the methods? Subaresas of what?

METHODS

135-138 – what does Fig. 1 have to do with this/ how does it support this sentence?

141 – might be worth providing some additional info on the delay-difference equations

155 – this is where you should introduce Fig 1. Otherwise it seems out of context, have no idea what subareas area etc

156-157 – management mechanism? Or ‘management relevant scale that actions can be implemented at’

157 – what do you mean by the entire model arena? (average of all models?) All 1001 trials? (which I only know about from later in the methods – needs to be explained at first mention)

160 – on what time scale? Plausible future for when?

174 – Can you provide further justification for only using RCP8.5 (I’m not disagreeing that you shouldn’t, but further justification should be provided)

185-187 – what link does estimating the breeding abundance of penguins using remote sensing have to do with developing a FBM? This needs to be made clearer to better justify the claim that both strategies have interest and support from various stakeholders.

188 – what are some of the alternative indicators/ possible approaches?

248 – what recent fishing patterns and from where?

RESULTS

278 – better worded as ‘Relative abundances of krill predators were sensitive to…’?

281 – so actively managing fisheries based on penguin abundance leads to a decline in penguins and seals (therefore managers do worse than if they did nothing) – is that realistic? Coming back to this after reading the discussion section about FBM-Pengs – you explain why the model might have produced the results it did, though do you think the model behaviour is realistic?

Fig 2. When you say ‘All results are referenced to the No FBM or No MPA scenarios.’ do you mean via the dotted line? (if so that should be specified)

Fig 3. Might be useful to have the timeframes labelled on the figure itself. For the broad overall trends presented in Fig 2 – do these only include the average of the SSMUs, or did you model the predator abundance across the subareas too? (can you specify)

Fig 5. Can you include the MPA’s in the figure – hard to compare results against it when it is not labelled/depicted. Also given that figures should stand alone without needing the text, you might need the full text in each figure legend (rather than referring to Fig 3) – but this probably depends on the journals standards.

Fig 6. Dashed line is ‘No FBM’? – you should add that info in (same comment as for Fig 2.)

DISCUSSION

375 – 381 – This is better suited to the introduction. The discussion should be giving the stated insights.

383 – simplify. Suggest something like ‘Decision makers must consider trade-offs in determining the best way to manage the krill industry’

390 – not sure this is an appropriate use of the word offset, at least in a conservation sense (is the decreased penguin abundance really adequately offset by a lower probability of fishing in areas of low krill density?)

393 – But only a single seal population declined (based on figure 5 and 6)?

391 & 394 – when you are talking about areas of low krill density, are you referring to the ‘threshold violation’ results from Fig 2? This should be made clearer in both the results and the discussion, else it appears that there are no results depicting low krill density and it is hard for the reader to find support for this statement (and either way there are no spatial results indicating where fishing occurred in low krill density areas?)

397 – 399 – round-about way of writing the sentence, please simplify

401 – 403 – Suggesting that managing Krill via FBM as a conservation strategy for predators is somewhat misleading – there were only extremely minor benefit to penguins using FBM-Krill and declines in seals (Figure 2) and there was no benefit to the fishery (Fig 2 & your statement in #299). Yet it would cost money to apply FBM and update it each time, and this is resources that could go to managing predator populations in a more effective way?

405 – What trade-offs? Please specify.

409 – 411 – Good, this is an important finding

440 – 445 – Good, this is a good conclusion

449 – potentially reword? ‘It is widely believed that routine application …’

458 – people? Fisheries is a better term

489 – Words like panacea and pathological can be simplified

516 – The sentence beginning with ‘In fact’ doesn’t make sense (before CCAMLR?)

520 – 522 – Sentence doesn’t make sense – something about the structure/position of the commas.

523 – instead of ‘steps’, something like ‘advancement toward’

REFERENCES

34/35 – you generally can’t base statements on things that haven’t been published or are not in press yet (see lines 118/119 – I went to look at the reference because I wanted to know more)? Are these in press? (what does forthcoming mean?)

Reviewer #2: It is difficult to fully evaluate this manuscript as essential methodological detail is missing. The authors cite earlier papers in lieu of providing these details but I don't think this appropriate for basic information that is essential to understanding what the authors have done and, hence, for evaluating the results and conclusions. This is why I have answered "Partly" and "No" to questions 1 and 2.

6. PLOS authors have the option to publish the peer review history of their article (what does this mean?). If published, this will include your full peer review and any attached files.

Reviewer #1: No

Reviewer #2: No

---

## [Author Response · Author response to Decision Letter 0]

30 Nov 2019

We thank the editor and the two reviewers for their comments on our initial submission of this manuscript. Overall, the comments have provided insightful and valuable feedback that has improved the manuscript, and we hope we have addressed their concerns. We have edited the manuscript to address concerns, and, as the comments and our specific responses are lengthy, have included these as a separate file in the revision.

---

## [Decision Letter · Decision Letter 1]

7 Jan 2020

PONE-D-19-22435R1

Comparing feedback and spatial approaches to advance ecosystem-based fisheries management in a changing Antarctic

PLOS ONE

Dear Dr Klein,

Thank you for submitting your revised manuscript to PLOS ONE. After consulting with one of the previous referees I would like to invite you to revise once more your paper so as to further improved your manuscript and get it ready for publication. I have selected the Minor Revision decision even though it may be slightly time consuming (hopefully not much) for you to address the final comments. I concur with the referee that it is necessary to give a bit of context to the Code that you've published on GitHub and that additional details on the Methodology are therefore needed to ensure that readers could use your approach without having to search other publications to do so. I would thus ask you to follow the recommendation of the referee to provide a bit of context in a Supplementary material so as not to make the reading too cumbersome. I understand that this entails extra work but, besides improving the chances of your paper to be used and cited, this would also allow for your manuscript to meet with the third Plos One criterion "Experiments, statistics, and other analyses are performed to a high technical standard and are described in sufficient detail."  

We would appreciate receiving your revised manuscript by Feb 21 2020 11:59PM. To enhance the reproducibility of your results, we recommend that if applicable you deposit your laboratory protocols in protocols.io, where a protocol can be assigned its own identifier (DOI) such that it can be cited independently in the future. For instructions see: http://journals.plos.org/plosone/s/submission-guidelines#loc-laboratory-protocols

We look forward to receiving your revised manuscript.

Kind regards,

Yan Ropert-Coudert, PhD

Academic Editor

PLOS ONE

Reviewers' comments:

Reviewer's Responses to Questions

**Comments to the Author**

1. If the authors have adequately addressed your comments raised in a previous round of review and you feel that this manuscript is now acceptable for publication, you may indicate that here to bypass the “Comments to the Author” section, enter your conflict of interest statement in the “Confidential to Editor” section, and submit your "Accept" recommendation.

Reviewer #2: (No Response)

2. Is the manuscript technically sound, and do the data support the conclusions?

Reviewer #2: Yes

3. Has the statistical analysis been performed appropriately and rigorously? 

Reviewer #2: Yes

4. Have the authors made all data underlying the findings in their manuscript fully available?

Reviewer #2: Yes

5. Is the manuscript presented in an intelligible fashion and written in standard English?

Reviewer #2: Yes

6. Review Comments to the Author

Reviewer #2: I appreciate the Authors' detailed response to my review and generally agree with their fundamental points regarding: 1) the length of additional context required to fully describe their previously published model; and 2) the citation of previous studies to support deriving studies/publications. Regarding point 2, I would add that any published study should be sufficiently self-contained that readers (and reviewers) can understand its methodology, at least conceptually, and therefore be able to judge the value of the results and conclusions drawn. I found some methodological context lacking in the original manuscript, and, as I stated in my original review, did not have access at that time to the preceding publications in order to better understand the model structure. In situations where essential aspects of a submitted paper's methodology are published elsewhere, Authors should provide the additional publication(s) as supporting documents to aid review and/or aggregate and condense those details in Supplementary Information. Both of these practices are commonplace and a courtesy to reviewers and interested readers by not placing upon them the burden of searching for the information across (potentially) multiple sources. Providing such detail as Supporting Information also enhances a study's reproducibility, which is also a fundamental element of publication (see my comment on line 168, below). Citation of the preceding studies, as the Authors stated, serves to assure readers that the foundational work has been vetted by peer review.

I do realise this approach places an extra burden on getting work published, at times I have grudgingly laboured under this burden myself, but it generally enhances readership and citation of numerically intensive studies.

Upon reading both the Author's Response to my original review and their revised manuscript, I am satisfied that most of the issues I originally raised have been addressed. Figure 2 nicely provides more context on the overall approach, and the discussion of model assumptions is a welcome addition.

A fairly minor point: given the two FBM strategies considered rely on updating catch limits based on changes in either krill density or penguin abundance, I still find it confusing that the Authors state that they "...focus on penguins and seals to simplify results..." (lines 152-153). I understand that the ecosystem model includes all the major krill predator groups but I can find no prior rationale why seals need be considered in these results when the FBM strategies are based on either krill or penguin indicators. Presumably, the predicted relative abundance of seals under the differing strategies provides a comparison to penguins, thereby giving a sense of how each strategy may benefit (or not) the broader ecosystem. An extra sentence in the last paragraph of the Introduction is probably all that is needed to clarify.

Specific Comments:

Figure 1 - it would be helpful to label the SSMU's 1-15, even if a smaller font is required. Doing so would then give spatial context to Table 1, which could be pointed out in the caption to Table 1.

Line 163 - have these data been published? if so, would be useful to cite.

Line 168 - I am pleased to see the code is now provided in a GitHub repository with a detailed description and analysis guide for readers interested in reproducing or extending the Authors' work. The Authors probably did not intend this but it's useful to point out that while such code repositories are enormously helpful for reproducibility, they are not a replacement for a sufficient explanation of a study's methodology.

7. PLOS authors have the option to publish the peer review history of their article (what does this mean?). If published, this will include your full peer review and any attached files.

Reviewer #2: No

---

## [Author Response · Author response to Decision Letter 1]

3 Apr 2020

This text has also been included as a separate document in the re-submission packet. 

Response to reviewer comments 

We thank the editor and the reviewer for their continued efforts with our work, and we hope we have addressed the final needs on this paper. We appreciate their time and support. 

Reviewer #1: 

Comment: I appreciate the Authors' detailed response to my review and generally agree with their fundamental points regarding: 1) the length of additional context required to fully describe their previously published model; and 2) the citation of previous studies to support deriving studies/publications. Regarding point 2, I would add that any published study should be sufficiently self-contained that readers (and reviewers) can understand its methodology, at least conceptually, and therefore be able to judge the value of the results and conclusions drawn. I found some methodological context lacking in the original manuscript, and, as I stated in my original review, did not have access at that time to the preceding publications in order to better understand the model structure. In situations where essential aspects of a submitted paper's methodology are published elsewhere, Authors should provide the additional publication(s) as supporting documents to aid review and/or aggregate and condense those details in Supplementary Information. Both of these practices are commonplace and a courtesy to reviewers and interested readers by not placing upon them the burden of searching for the information across (potentially) multiple sources. Providing such detail as Supporting Information also enhances a study's reproducibility, which is also a fundamental element of publication (see my comment on line 168, below). Citation of the preceding studies, as the Authors stated, serves to assure readers that the foundational work has been vetted by peer review.

I do realise this approach places an extra burden on getting work published, at times I have grudgingly laboured under this burden myself, but it generally enhances readership and citation of numerically intensive studies.

Upon reading both the Author's Response to my original review and their revised manuscript, I am satisfied that most of the issues 

I originally raised have been addressed. Figure 2 nicely provides more context on the overall approach, and the discussion of model assumptions is a welcome addition.

Response: We again thank this reviewer for their time and for continuing to engage with us to improve this manuscript. We also agree with their points here, and the importance of manuscripts standing alone – the discussion here illuminates, at least to us, a challenge in publication that should be more deeply considered, and we thank this Reviewer for urging us to do so. We have added additional text to the manuscript and Supporting Information, as well as two tables and READ ME files to the SI. We are hopeful this will work; if additional details are necessary, we ask for more specific instruction on what would be useful. We sincerely wish to adequately address these concerns but it remains somewhat unclear to us what is needed as well as also possible without recreating the previously published text of Watters et al. (2013). 

Comment: A fairly minor point: given the two FBM strategies considered rely on updating catch limits based on changes in either krill density or penguin abundance, I still find it confusing that the Authors state that they "...focus on penguins and seals to simplify results..." (lines 152-153). I understand that the ecosystem model includes all the major krill predator groups but I can find no prior rationale why seals need be considered in these results when the FBM strategies are based on either krill or penguin indicators. Presumably, the predicted relative abundance of seals under the differing strategies provides a comparison to penguins, thereby giving a sense of how each strategy may benefit (or not) the broader ecosystem. An extra sentence in the last paragraph of the Introduction is probably all that is needed to clarify.

Response: We find this comment quite important, and critical that we clarify. First, the Reviewer is correct in their assessment: indicators are useful as they can simplify management objectives and approach, but there will still be larger, ecosystem outcomes of changes in management beyond the indicator species (of course). Here, outcomes for seals allow us some limited insight into how using one indicator species may have implications for others – i.e. comparing consequences for seals to those for penguins, as the Reviewer surmises. Second, our reasoning for seals and penguins in the main text and having other krill predators in Supporting is not to say these other species are less important or are not also a window into impacts on additional outcomes, but to maintain a manageable set of results in the text, given our figures are already extensive with two species – i.e. that is a logistical consideration only. 

It is important these points are clear for readers, and we have addressed the text to hopefully make them so (starting line 142). 

Comment: Figure 1 - it would be helpful to label the SSMU's 1-15, even if a smaller font is required. Doing so would then give spatial context to Table 1, which could be pointed out in the caption to Table 1.

Response: We have altered Figure 1, increasing the map size and using red to help the SSMU numbers stand out. We have also noted it in the caption for Table 1 as helpfully suggested.

Comment: Line 163 - have these data been published? if so, would be useful to cite.

Response: We believe this comment refers to the reference set of four parameterization, which is indeed published and the citation is referenced (Watters et al., reference 39 in the main text). 

Comment: Line 168 - I am pleased to see the code is now provided in a GitHub repository with a detailed description and analysis guide for readers interested in reproducing or extending the Authors' work. The Authors probably did not intend this but it's useful to point out that while such code repositories are enormously helpful for reproducibility, they are not a replacement for a sufficient explanation of a study's methodology.

Response: We agree, and have added additional text in the main manuscript and Supporting Information to this effect.

---

## [Editor Report · Decision Letter 2]

6 Apr 2020

Comparing feedback and spatial approaches to advance ecosystem-based fisheries management in a changing Antarctic

PONE-D-19-22435R2

Dear Dr. Klein,

We are pleased to inform you that your manuscript has been judged scientifically suitable for publication and will be formally accepted for publication once it complies with all outstanding technical requirements.

With kind regards,

Yan Ropert-Coudert, PhD

Academic Editor

PLOS ONE
---

## [Editor Report · Acceptance letter]

26 Aug 2020

PONE-D-19-22435R2 

Comparing feedback and spatial approaches to advance ecosystem-based fisheries management in a changing Antarctic 

Dear Dr. Klein:

I'm pleased to inform you that your manuscript has been deemed suitable for publication in PLOS ONE. Congratulations! Your manuscript is now with our production department. 

Kind regards, 

on behalf of

Dr. Yan Ropert-Coudert 

Academic Editor

PLOS ONE